# Genome-Scale Metabolic Model of the Human Pathogen *Candida albicans*: A Promising Platform for Drug Target Prediction

**DOI:** 10.3390/jof6030171

**Published:** 2020-09-11

**Authors:** Romeu Viana, Oscar Dias, Davide Lagoa, Mónica Galocha, Isabel Rocha, Miguel Cacho Teixeira

**Affiliations:** 1Department of Bioengineering, Instituto Superior Técnico, Universidade de Lisboa, 1049-001 Lisbon, Portugal; romeuviana@tecnico.ulisboa.pt (R.V.); monicagalocha@gmail.com (M.G.); 2Institute for Bioengineering and Biosciences, Biological Sciences Research Group, Instituto Superior Técnico, 1049-001 Lisbon, Portugal; 3Centre of Biological Engineering, Universidade do Minho, 4710-057 Braga, Portugal; odias@ceb.uminho.pt (O.D.); dlagoa@ceb.uminho.pt (D.L.); 4Instituto de Tecnologia Química e Biológica António Xavier, Universidade Nova de Lisboa (ITQB-NOVA), 2780-157 Oeiras, Portugal

**Keywords:** *Candida albicans*, global stoichiometric model, drug targets, metabolic reconstruction, gene essentiality

## Abstract

*Candida albicans* is one of the most impactful fungal pathogens and the most common cause of invasive candidiasis, which is associated with very high mortality rates. With the rise in the frequency of multidrug-resistant clinical isolates, the identification of new drug targets and new drugs is crucial in overcoming the increase in therapeutic failure. In this study, the first validated genome-scale metabolic model for *Candida albicans*, iRV781, is presented. The model consists of 1221 reactions, 926 metabolites, 781 genes, and four compartments. This model was reconstructed using the open-source software tool *merlin* 4.0.2. It is provided in the well-established systems biology markup language (SBML) format, thus, being usable in most metabolic engineering platforms, such as OptFlux or COBRA. The model was validated, proving accurate when predicting the capability of utilizing different carbon and nitrogen sources when compared to experimental data. Finally, this genome-scale metabolic reconstruction was tested as a platform for the identification of drug targets, through the comparison between known drug targets and the prediction of gene essentiality in conditions mimicking the human host. Altogether, this model provides a promising platform for global elucidation of the metabolic potential of *C. albicans*, possibly guiding the identification of new drug targets to tackle human candidiasis.

## 1. Introduction

In the last few decades, a significant increase in nosocomial fungal infections has been observed, and *Candida* species are by far the most common cause of invasive fungemia in humans [1,2]. Among *Candida* species, *Candida albicans* is the main etiological agent of invasive candidiasis [3,4], being associated to high mortality rates [4]. Together with its virulence traits [5,6], its ability to acquire drug resistance [7,8,9] makes this opportunistic pathogen a severe threat.

Only three classes of antifungal drugs are licensed to treat *Candida* infections (azoles, echinocandins, and amphotericin B), and only some azoles and echinocandins are recommended as first-line agents [10]. Currently, there has been a rise in the frequency of multidrug-resistant clinical isolates, and therapeutic options are running low. This is true for *C. albicans*, but even more so for other emerging non-albicans *Candida* species, such as *C. glabrata*, *C. krusei* and *C. auris*. For example, in recent studies, almost 40% of the *Candida glabrata* isolates shown to be resistant to at least one echinocandin were also resistant to fluconazole [11,12]. In non-*albicans* pathogenic *Candida* species, the scenario is even more frightening, as several of them display, either intrinsic or easily acquired resistance to several of the available antifungal agents. For example, in a recent case, *Candida auris* isolates were identified as resistant to the three classes of available antifungal drugs, further raising public concern on the future efficacy of current antifungal therapeutic options [13]. The identification of new drug targets and new drugs is crucial to overcome the increase in therapeutic failure.

Genome-scale metabolic models have the potential to provide a holistic view of cell metabolism. Historically, these global mathematical descriptions of cell metabolism have mostly been linked to metabolic engineering of microbial cell factories, given their potential to simulate global metabolic behavior and provide hints to guide experimental optimization of such organisms for the production of added-value compounds [14]. However, recent examples have shown the potential of these models in the quest for novel drug targets in pathogenic organisms [15,16,17,18,19]. For example, Abdel-Haleem et al. in 2018, described the reconstruction of genome-scale metabolic models for five life cycle stages of *Plasmodium falciparum*, enabling the identification of potential drug targets that could be used as both, anti-malarial drugs and transmission-blocking agents [20].

Here, we present the first validated in silico genome-scale metabolic reconstruction of *C. albicans*, the iRV781. This model is provided in the well-established SBML format and can easily be read in most metabolic engineering platforms such as OptFlux [21] and COBRA [22]. The model validation procedure is detailed, and evaluation of the potential of this model for research is advanced for new drug targets in this fungal pathogen.

## 2. Materials and Methods

### 2.1. Model Development

The *Candida albicans* iRV781 genome-scale metabolic model was developed following the methodology represented in Figure 1, using *merlin* 4.0.2 [23] for the reconstruction process, as described elsewhere [24], and OptFlux 3.0 [21], for the curation and validation of the model. All predictions were performed using the IBM CPLEX solver (IBM, Armonk, NY, USA). *Merlin* is a platform that enables the semi-automatic reconstruction of metabolic models, providing a user-friendly interface that assists the user in the manual curation process [19].

### 2.2. Genome Annotation and Assembling the Metabolic Network

The genome sequence of the reference strain *Candida albicans* SC5314 was obtained from NCBI’s Assembly database, accession number ASM18296v3 (www.ncbi.nlm.nih.gov/assembly) [25] and the Taxonomy ID from NCBI (www.ncbi.nlm.nih.gov/taxonomy) [26], which is required by *merlin* to univocally identify the organism under study throughout the reconstruction process. In order to establish a proximity between species, the 16S rRNA gene of several known closely related species was used to construct a Phylogenetic tree, the sequences being retrieved from NCBI’s database and aligned using MEGA X 10.0.5 (Pennsylvania State University, State College, PA, USA) [27]. The evolutionary history was inferred by using the Maximum Likelihood method and Tamura-Nei model [28] (Appendix A). The genome-wide functional annotation was processed by *merlin* based on taxonomy and frequency of similar sequences trough remote Basic Local Alignment Search Tool (BLAST) [29] similarity searches to the UniProtKB/Swiss-Prot database [30] (http://www.UniProt.org/) and HMMER [31]. Protein-reaction associations, available in the Kyoto Encyclopedia for Genes and Genomes (KEGG) BRITE database, were used to assemble the draft network. All reactions classified as spontaneous or non-enzymatic were also included in the first draft of the model. The assembly of the metabolic network is performed by *merlin*, using genome annotation to determine which reactions will be included in the model, based on an algorithm described in detail elsewhere [23].

### 2.3. Reversibility and Balancing

In order to ensure that all reactions in the network are balanced, unbalanced reactions were identified, using the corresponding *merlin* tool, manually verified and corrected. Reaction reversibility was also confirmed to avoid gaps and mispredictions of the model, through the corresponding *merlin* tool and using Braunschweig Enzyme Database (BRENDA) [32] as reference and the data provided elsewhere [33]. Since there are no guarantees that the EC numbers available in the different databases will be updated, a manual inspection was also performed to correct a few cases of enzymes with deleted/transferred EC numbers, using public databases (BRENDA [34], UniProt, MetaCyc [35] and KEGG [36]) and literature search.

### 2.4. Compartmentalization

This model includes four compartments: Extracellular, cytoplasm, mitochondrion and cytoplasmic membrane. The prediction of compartments for each enzyme and carrier was performed using the WoLF PSORT protein subcellular localization predictor [37].

### 2.5. Transport Reactions

Given the existence of compartments in the model, it is necessary to create transport reactions for the metabolites. Transport reactions were generated using genomic information together with the public database TCDB [35] by *merlin*’s TranSyT [36]. Transport reactions across internal and external membranes for currency metabolites, such as H_2_O, CO_2_, and NH_3_, which are often carried by facilitated diffusion, were added to the model with no gene association.

### 2.6. Biomass Equation

The biomass formation was represented by an equation that includes proteins, DNA, RNA, lipids, carbohydrates, and cofactors, and detailed information for the composition of each one of these macromolecules. The content of each component was determined based on the literature or using experimental data. All the calculations were performed as described previously [38]

For the phosphorus to oxygen ratio the same theoretical ratio used in the *S. cerevisiae* iMM904 metabolic model was applied, 1.5.This ratio represents the relationship between ATP synthesis and oxygen consumption, indicating the number of orthophosphate molecules used for ATP synthesis per atom of oxygen consumed during oxidative phosphorylation [14]. Three generic reactions contributing to this ratio were automatically generated by *merlin*, and were updated to replicate the same ratio as in the iMM904 model:

Reaction R00081_C4:1.0 Oxygen_mito_ + 4.0 Ferrocytochrome c_mito_ + 6.0 H^+^_mito_ ↔ 2.0 H2O_mito_ + 4.0 Ferricytochrome c_mito_ + 6.0 H^+^_cyto_(1)

Reaction T02161_C4:1.0 Ubiquinol_mito_ + 2.0 Ferricytochrome c_mito_ + 1.5 H^+^_mito_ ↔ 1.0 Ubiquinone_mito_ + 2.0 Ferrocytochrome c_mito_ + 1.5 H^+^_cyto_(2)

Reaction T00485_C4:1.0 Orthophosphate_mito_ + 1.0 ADP_mito_ + 3.0 H^+^_cyto_ ↔ 1.0 ATP_mito_ + 1.0 H2O_mito_ + 3.0 H^+^_mito_(3)

The final balance reaction:3.0 Orthophosphate_mito_ + 1.0 Oxygen_mito_ + 3.0 ADP_mito_ + 2.0 Ubiquinool_mito_ ↔ 3.0 ATP_mito_ + 5.0 H_2_O_mito_ + 2.0 Ubiquinone_mito_(4)

This model also includes ATP requirements for biomass formation and maintenance (non-growth). The growth ATP requirements, 23.346 mmoles ATP/g Dry Cell Weight (DCW), were introduced directly into the biomass equation; this value was calculated based on ATP requirements for biosynthesis of cell polymers for *S. cerevisiae*, adjusted for the composition in macromolecules of the biomass equation [39].

Non-growth associated ATP maintenance, the amount of ATP required by the cell even when it is not growing, was represented in the model by an equation that forces ATP consumption via a specific flux. The boundaries of this flux were inferred from *Candida tropicalis* [40]. See Appendix A for more detailed information on the computation of the biomass equation.

### 2.7. Curation of the Model

Throughout the curation process, reactions were edited, manually added to, or removed from the model to correct some gaps in the network, using KEGG pathways, MetaCyc Database, and literature data as standards.

### 2.8. Strains and Growth Media

*Candida albicans* reference strain SC5314 was batch-cultured at 37 °C, with orbital agitation (250 rpm) in Yeast Nitrogen Base (YNB) medium without amino acids: 5 g/L glucose (Merck), 6.8 g/L YNB (Difco). Solid media contained, besides the above-indicated ingredients, 20 g/L agar (Iberagar).

### 2.9. Carbon and Nitrogen Source Utilization Assessment

The capability of utilizing different carbon and nitrogen sources for cell growth was assessed by comparing in silico predictions to literature data for *C. albicans*. For the few carbon or nitrogen sources for which the model predictions were not consistent with literature data, wet-lab experiments were conducted. Specifically, the utilization of cellobiose, D-Ribose, and mannitol as carbon source, by the *C. albicans* reference strain SC5314, was evaluated in solid YNB medium containing either 5 g/L glucose as control, or 5 g/L of either one of the mentioned carbon sources. *C. albicans* cell suspensions used to inoculate the agar plates, were mid-exponential cells grown in YNB medium with 5 g/L glucose, until culture OD600nm = 0.5 ± 0.05 was reached and then diluted in sterile water to obtain suspensions with OD600nm = 0.05 ± 0.005. These cell suspensions and subsequent dilutions (10^−1^; 10^−2^; 10^−3^) were applied as 4 µL spots onto the surface of solid YNB media, with the indicated carbon sources. Growth was assessed after incubation at 37 °C for 24 h.

### 2.10. Network Simulation and Analysis

All the phenotype simulations were performed with Flux Balance Analysis (FBA) in OptFlux 3.0 [21] using the IBM CPLEX solver (IBM, Armonk, NY, USA). Gene essentiality was also determined by OptFlux 3.0 which provides a tool that allows to determine critical genes automatically by performing individual gene knockouts and simulating growth in a given environmental condition. Environmental conditions that simulated the Roswell Park Memorial Institute (RPMI, Buffalo, NY, USA) medium were used, in order to replicate the human serum conditions.

## 3. Results and Discussion

### 3.1. Model Characteristics

The final version of the iRV781 model includes 781 genes associated with 1221 reactions, among which, 174 are transport reactions, and 196 are external drain reactions (exchange constraints set to mimic the environmental conditions), involving 927 metabolites and four different compartments. Analyzing the distribution of proteins by compartments, 205 are plasma membrane proteins, 521 cytoplasmatic proteins and 139 mitochondrial proteins.

In order to elucidate the characteristics of our model we selected well-characterized genome-scale metabolic models of *C. glabrata* [41] and *S. cerevisiae* [42] as a comparison. Table 1 shows the distribution of those reactions by the main pathways in the three models. In general, the number of reactions by pathway is quite similar to *C. glabrata*, *S. cerevisiae* or both.

Although our model has common standard identifiers for reactions (KEGG ID), it is not possible to assess how the reactions differ among the three models, since the remaining two models do not possess the same identifiers. However, considering only the proteins associated with an EC number, it is possible to make a comparison across the existing models. More than 80% of the proteins with an associated EC number in our model are also present at least in one of the other 2 models (*S. cerevisiae* or *C. glabrata*). Furthermore, about 65% of the proteins are shared by the three models while about 20% are unique in iRV781 (Figure 2). The complete list of unique EC numbers can be found in Appendix A.

In most cases, the observable differences in EC numbers were related to outdated EC numbers or were compensated with other enzymes that are responsible for the same reactions in the model. However, some cases stand out as potential unique features of *C. albicans*:The enzyme 1.13.99.1, inositol oxygenase, responsible for the conversion of myo-inositol into D-glucuronate. This enzyme seems to be involved in resistance to toxic ergosterol analogs [44], is also present in other *Candida* species, including some important pathogens (*C. parapsilosis*, *C. dubliniensis*, *C. auris)*, but absent in *C. glabrata*.The enzyme 1.1.1.289, sorbose reductase, responsible for the interconversion of L-sorbose into D-sorbitol. In fact, the presence this enzyme allows *C. albicans* to use L-sorbose as carbon source [45], unlike *S. cerevisiae*.The enzyme 1.14.19.17, sphingolipid 4-desaturase, responsible for the conversion of dihydroceramide into *N*-Acylsphingosine. This protein is involved in sphingolipid metabolism, with possible impact in azole resistance in *C. albicans* [46]. The presence of this enzyme may represent a specific resistance feature of some *Candida* species, being present in *C. parapsilosis*, *C. dubliniensis*, *C. auris,* but not in *C. glabrata*.The enzyme 1.1.99.2, L-2-hydroxyglutarate dehydrogenase, is a metabolite repair enzyme responsible for the conversion of (S)-2-hydroxyglutarate into 2-oxoglutarate. In other organisms such as plants [47] or humans [48], the inactivation of this enzyme leads to the accumulation of the toxic (S)-2-hydroxyglutarate.The enzyme 2.7.1.59, *N*-acetylglucosamine kinase, responsible for the conversion of *N*-acetyl-D-glucosamine into *N*-acetyl-D-glucosamine 6-phosphate. Many yeast species, including *S. cerevisiae* have lost their ability to utilize *N*-acetyl-D-glucosamine as carbon source, however, genetically altered yeasts are able to use it, based on expression of *C. albicans* genes [49]. In fact, this enzyme allows *C. albicans* to utilize this carbon source, a feature that is particularly important for its survival inside the phagosomes [50].The enzyme 3.5.1.25, *N*-acetylglucosamine-6-phosphate deacetylase, responsible for the conversion of *N*-acetyl-D-glucosamine 6-phosphate into D-glucosamine 6-phosphate. Like 2.7.1.59, this enzyme is also involved in *N*-acetyl-D-glucosamine metabolism.The enzyme 1.4.3.3, D-amino-acid oxidase, responsible for the conversion of a D-amino acid into a 2-oxo carboxylate and ammonia, is the first enzyme involved in the catabolism of D-amino acids and may allow the utilization D-amino acids as a source of carbon or nitrogen in some yeasts [51]. It may be an interesting feature to be explored in *C. albicans*.

#### 3.1.1. Gap Filling and Model Curation

During the process of manual curation described in the methods section, a total of 66 reactions were manually added to the initial model obtained from the results of re-annotation to fill gaps. Additionally, evidence from the literature was always considered, or of the well-studied *S. cerevisiae*. On the other hand, 336 reactions were removed from the initial model, have been removed for being unconnected reactions, general reactions, reactions using metabolites that are not included in the model, or reactions for which it was manually verified that the model does not have the enzyme coding gene. Additionally, the compartment of 79 reactions was changed, and 94 reactions were altered to become balanced. The complete list of alterations can be found in Appendix A.

#### 3.1.2. Biomass Equation

The biomass equation (Table 2) includes the composition of proteins, DNA, RNA, lipids, carbohydrates, and cofactors. For the composition of DNA, the whole genome sequence was used to estimate the amount of each deoxyribonucleotide as described in [52], while mRNA, rRNA, and tRNA were used to estimate the total RNA in the cell as described in [14]. For the amino acid composition, the percentage of each codon usage was calculated from the translated genome sequence [52], using the e-BiomassX tool [53].

Carbohydrate [54], Lipid [55], Sterol [55], Phospholipid [56], and Fatty acid [57] compositions were inferred from literature data. Essential metabolites were included in the biomass composition to qualitatively account for the essentiality of their synthesis pathways [41,58]. The growth and non-growth ATP requirements were adopted from *S. cerevisiae* [59].

### 3.2. Validation of the iRV781 Model

#### 3.2.1. Carbon and Nitrogen Source Utilization

Based on the literature, phenotypic growth data were collected from different sources. Data related to *C. albicans* strains, other than the reference SC5314 strain, was also considered in the analysis to increase the number of carbon and nitrogen sources tested.

In a first simulation, this model correctly predicted the usability of 92% of the 39 tested carbon sources. According to data available on Royal Netherlands Academy of Arts and Sciences (CBS-KNAW) Fungal Biodiversity Centre webpage [60], the *C. albicans CBS562* strain seems not to be able to use cellobiose or D-ribose as sole carbon sources, contrary to the model’s prediction. Therefore, the utilization of cellobiose and D-ribose by *C. albicans* SC5314 was evaluated experimentally to assess whether the prediction failure could result from a different metabolic capacity exhibited by the reference strain. The results confirmed the model’s prediction regarding the utilization of cellobiose and D-Ribose (Figure 3), suggesting that the reference *C. albicans* strain has higher metabolic capabilities, when compared to other strains. The model’s prediction failed only for mannitol that, according to the model, cannot be used as sole carbon source by *C. albicans*, contradicting experimental evidence gathered for the *C. albicans* SC5314 strain [61,62] (Figure 3). It was not possible to identify the source of this problem in the built model. Nonetheless, the model is able to correctly predict the usability of 97% of the tested substrates.

Altogether, the constructed model proved accurate when predicting the utilization of different carbon and nitrogen sources, when compared to experimental data (Table 3). It correctly predicts the usability of 97% of the tested carbon sources, and 80% of the 15 tested nitrogen sources. It should be noted that in the nitrogen tests, none of the literature data was obtained using the reference strain; therefore, it is likely that the *C. albicans* SC5314 strain is able to use a larger number of nitrogen sources than the previously tested strains.

#### 3.2.2. Growth Parameters in Batch Culture

Experimental data obtained elsewhere [65] from synthetic minimal media batch cultures with glucose as carbon source were used to validate the model quantitatively. The model was simulated in environmental conditions that simulate the medium used in Rozpȩdowska et al., 2011. The glucose uptake flux was fixed to q_Glucose_ = 7.56 mmol g^−1^ dry weight h^−1^ as per such work, and the remaining nutrients flux were left unconstrained, as the model in this condition is glucose-limited. Once again, the model proved to be robust as the experimentally observed growth rate is similar to that predicted by the model (Table 4). Additionally, the formation of glycerol, acetic acid, and ethanol as by-products was not predicted to occur, which is in agreement with the experimental data, except for ethanol, that appears to be produced in trace amounts. *C. albicans*, as a crabtree-negative yeast [66], under aerobic conditions does not produce significant concentrations of ethanol. Nonetheless, the model predicts ethanol production under low-oxygen conditions (q_Oxygen_ < 7.56 mmol g^−1^ dry weight h^−1^).

*C. albicans* is unable to grow in anaerobic conditions in minimal media. However, *C. albicans* colonization is known to spread into anaerobic niches of the gastrointestinal tract or in the inner sections of biofilms where the oxygen availability is scarce or null. Dumitru et al., 2004 reported a defined anaerobic growth medium for studying *Candida albicans*. In this medium (GPP) oleic acid and nicotinic acid were added as required growth factors for anaerobic growth [67]. Interestingly, *S. cerevisiae* under anaerobiosis also requires growth factors, such as ergosterol and Tween 80, a source of oleic acid, if growing in a defined medium (SMM), it should be noticed that SMM medium also contains nicotinic acid in its composition despite not being a required growth factor in anaerobic conditions [68]. Cultivation was simulated in the absence of oxygen in GPP medium and GPP medium supplemented with oleic acid and nicotinic acid, to assess if this model can predict growth in anaerobic conditions. Our model predicts the growth only in media supplemented with specific anaerobic growth factors. For the simulation, the glucose uptake flux was set to q_Glucose_ = 6.58 mmol g^−1^ dry weight h^−1^, to compare the growth parameters with the reported values for *S. cerevisiae*. Indeed, for the same anaerobic conditions, the specific growth rate of the model and the ethanol production are similar to data reported for *S. cerevisiae* [68], though the model does not predict the production of glycerol in such conditions (Table 5).

### 3.3. Gene Essentiality Assessment: A Tool for Drug Target Discovery?

A set of *C. albicans* essential genes [69], were used to evaluate the model’s ability to predict essentiality. For each gene a simulation was performed, on the same environmental conditions, described in the reference [69] (YNB medium), eliminating the corresponding reactions for that gene. Only protein-coding genes present in the model were considered. The model was able to correctly predict 78% (84 out of 108) of the identified essential genes (Appendix A). It is important to highlight that, in this type of models, the regulatory network is not considered, so it is expected that some predictions are not close to reality.

To evaluate whether the assessment of gene essentiality could be a promising tool in drug target discovery, each one of the identified essential enzymes in the RPMI medium was searched in the DrugBank database [70], as a possible drug target of known antimicrobial agents. RPMI medium simulates human serum, thus allowing to simulate the natural environment faced by *Candida albicans* in systemic infections.

Interestingly, 11 ERG genes, including the well-known azole drug target *ERG11*, were predicted by the model to be essential in RPMI medium. Although most ERG genes are not essential, the inhibition of the activity of this pathway has a fungistatic effect indeed. They encode the enzymes that guide the last steps of ergosterol biosynthesis. This pathway is the main target of azole drugs, one of the most common antifungal agents to treat *Candida* infections [71]. These drugs act by blocking ergosterol biosynthesis inhibiting the Erg11 encoded by the *ERG11* gene. When an azole drug binds to this enzyme, ergosterol synthesis is inhibited, leading to lower concentrations of this metabolite in the plasma membrane [72]. Given that ergosterol is part of the *C. albicans*’ biomass, it is acceptable to consider that enzymes that participate in its synthesis pathway can be essential, making most ERG genes attractive alternatives as new drug targets [73].

Many additional proteins stand out as promising new drug targets, including some for which there are already predicted inhibitory drugs, based on the results for homologous proteins in other organisms. For example, Atovaquone is a drug used as a fixed-dose combination with Malarone for treating uncomplicated malaria cases or as chemoprophylaxis in travelers. This drug is an analogue of ubiquinone and targets enzyme 1.3.5.2 encoded by *URA9* in *Plasmodium falciparum*. Atovaquone acts as a competitive inhibitor of ubiquinol inhibiting the mitochondrial electron transport chain at the bc1 complex, resulting in a loss of mitochondrial function [74]. It would be interesting to check if these drugs are also active against *C. albicans* by targeting CaUra9.

Another promising example of a predicted *C. albicans* drug target is Fol1, which corresponds to enzyme 2.5.1.15. Fol1 is the target of the sulfa drugs (sulfonamides and sulfones), a very well-known class of drugs, used to treat infectious diseases [75]. The effect of sulfa drugs on *C. albicans* has not been sufficiently investigated. However, it seems that sulfa-fluconazole combination results in increased antifungal activity against *C. albicans*, leading to the reversal of azole resistance in previously resistant strains [76].

Since a reaction can be catalyzed by a protein encoded by more than one gene, and genes may encode more than one protein, we decided to analyze the model’s critical reactions. This analysis allowed to increase the number of confirmed drug targets predicted as essential enzymes.

As an example, the FKS genes stand out. They are not considered essential as the enzyme beta-1,3-glucan synthase can be encoded by more than one FKS/GSC/GSL genes [77]. However, the model predicts the reaction in which this enzyme participates as essential. The beta-1,3-glucan synthase is the target of the echinocandin class of antifungal drugs. Via noncompetitive inhibition, these drugs block the enzyme and stop beta-1,3-glucan synthesis, compromising the integrity of the cell wall [78].

Other drugs, such as ethionamide, sulfacetamide, azelaic acid, cerulenin or trimethoprim, were identified as targeting proteins from various organisms. These proteins are homologous to the proteins encoded by genes identified as essential in RPMI, in the *C. albicans’* model (Table 6). Altogether, these results suggest that the iRV781 model may prove useful in the prediction of new drug targets. The predictions of this model may extend to other pathogenic *Candida* species. In fact, if we search the 12 genes present in Table 6 in emerging non-*albicans Candida* species, they all have orthologous genes in *C. parapsilosis* and *C. dubliniensis*, 11 of them have orthologs in *C. auris* and 8 of them in *C. glabrata*.

Despite the methodology for the reconstruction of genome-scale metabolic models being standardized, eukaryotic models remain a challenge, due to their large genomes and complexity [79]. These models always seek to get as close as possible to reality; however, given the complexity of the networks, they are always subject to some errors, which may cause small deviations in the predictions. Some errors may include incorrect assignment of GPR associations, reaction directionality or reversibility, incongruous stoichiometric parameters, missing reactions and inaccurate biomass composition [79]. Additionally, network properties, that go beyond metabolism, cannot be addressed with currently existing modeling tools at a global scale, thus limiting the predictive power that may be drawn from global stoichiometric models. Still, they provide a fresh view of a pathogen’s metabolism, while offering a tool to inspect the metabolism itself as a target for new drugs.

Genome-scale metabolic reconstructions are effective in drug target prediction and are expected to continue to expand in the future [80]. Gene essentiality assessment is the most common method to identify potential drug targets, and for a better prediction, it is necessary to consider the medium in which the organism is exposed. In this work, gene essentiality was searched for in RPMI medium in order to simulate the natural environment faced by *Candida albicans* in systemic infections. However, it is important to highlight that in theses reconstructions, it is not considered that cells may need time to adapt to genetic perturbations or environmental variability [79]. Additionally, yeast interactions with other microorganisms and the secretion of compounds that can influence their surrounding environment are not taken into account [81]. Despite these inaccuracies, genome-scale metabolic reconstructions have proved to be very efficient discovering new drug targets, and once a model is built, drug targets can be predicted relatively easily. In fact, the experimental validation of the targets and the identification of the effective drugs represents a more demanding challenge [82].

## 4. Conclusions

The first validated global metabolic model for the human pathogen *C. albicans* is presented in this study. The model was manually curated and validated thoroughly, constituting a powerful platform for the study of *C. albicans* metabolic potential and weaknesses. The iRV781 model includes 781 genes associated with 1221 reactions, the number of reactions in the main pathways being similar to those in *C. glabrata* and *S. cerevisiae* models. However, about 20% of the proteins associated with EC numbers in iRV781 are unique in relation to these models. The model proved accurate when predicting the utilization of different carbon and nitrogen sources, and in anaerobic growth in defined anaerobic media. In silico growth parameters are also in agreement with the experimental data. We were able to identifyas essential genes in RPMI medium some which are already known targets antifungal agents and other antimicrobial agents used in clinical practice. This observation suggests that the *C. albicans* global stoichiometric model, presented herein, may be a promising platform for the identification of further targets for new antifungal drugs.

## Figures and Tables

**Figure 1 jof-06-00171-f001:**
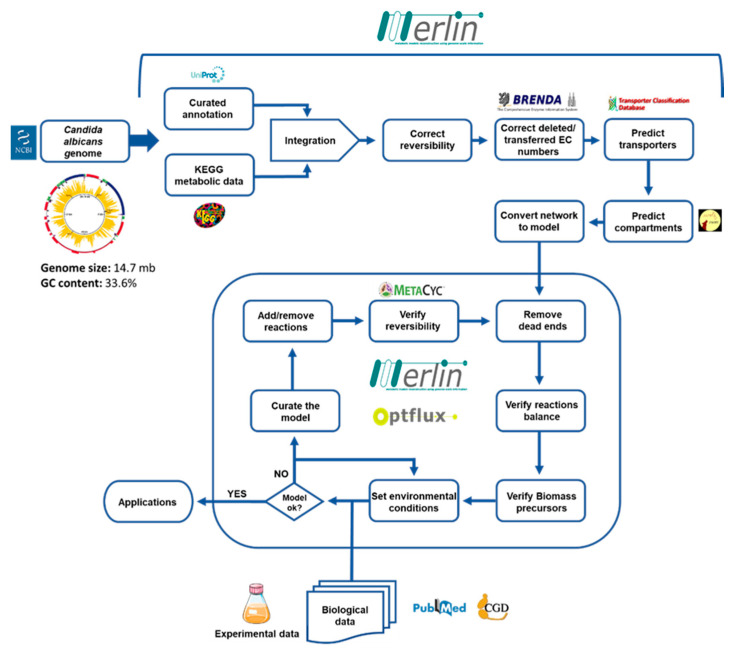
Methodology for the reconstruction of the *Candida albicans* iRV781 metabolic model. Adapted from [14].

**Figure 2 jof-06-00171-f002:**
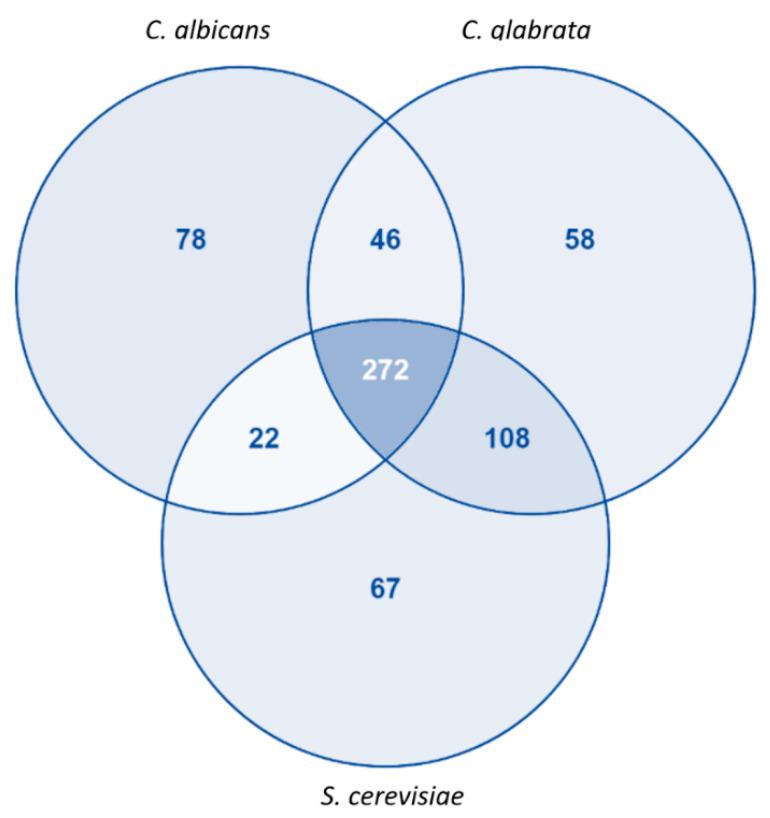
Comparison between *C. albicans*, *S. cerevisiae* and *C. glabrata* proteins with associated EC Numbers present in the genome-scale metabolic models iRV781, iIN800, and iNX804, respectively. Diagram obtained using VENNY2.1 tool [43].

**Figure 3 jof-06-00171-f003:**
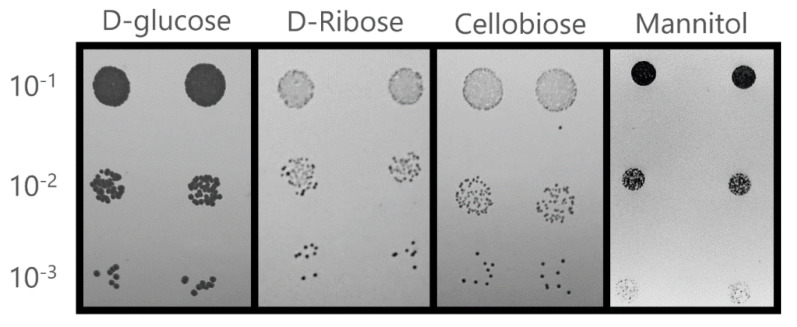
Utilization of glucose (control), cellobiose, D-Ribose, and mannitol by *C. albicans* reference strain SC5314 as carbon source in solid YNB medium. Initial OD600nm = 0.5 ± 0.05. Growth was assessed after incubation at 37 °C for 24 h.

**Table 1 jof-06-00171-t001:** Number of reactions in the main pathways of the *C. albicans* iRV781 model in comparison to C. *glabrata* iNX804 model and *S. cerevisiae* iMM904 model.

	*C. albicans*	*C. glabrata*	*S. cerevisiae*
	iRV781	iNX804	iMM904
Amino acid metabolism	218	223	217
NAD biosynthesis	20	20	24
Cofactors and vitamins	122	120	127
Nucleotide metabolism	120	138	135
Alternate carbon metabolism	27	31	27
Glycolysis/gluconeogenesis	26	18	22
Citrate cycle	24	20	13
Pentose phosphate pathway	18	16	13
Pyruvate metabolism	31	28	18
Oxidative phosphorylation	10	13	19
Sterol metabolism	29	30	49
Fatty acid metabolism	87	81	108
Glycerolipid metabolism	13	9	12
Phospholipid metabolism	34	44	52

**Table 2 jof-06-00171-t002:** Biomass Composition used in the model iRV781. More detailed information is found in Appendix A.

Metabolite	g/gDCW	Metabolite	g/gDCW
**Protein components**		**Lipids**	
L-Valine	0.02001	Lanosterol	0.00166
L-Tyrosine	0.02153	Squalene	0.00088
L-Tryptophan	0.00671	Ergosterol	0.00247
L-Threonine	0.02311	Phosphatidylserine	0.00299
L-Serine	0.02908	Phosphatidylinositol	0.00417
L-Proline	0.01616	Phosphatidylcholine	0.00681
L-Phenylalanine	0.02407	Phosphatidylethanolamine	0.00542
L-Methionine	0.00869	Cardiolipin	0.00201
L-Lysine	0.03535	Phosphatidic acid	0.00271
L-Leucine	0.03874	Phosphatidylglycerol	0.00174
L-Isoleucine	0.02992	Tetradecanoic acid	0.00003
L-Histidine	0.01067	Hexadecanoic acid	0.00073
L-Glutamate	0.03084	Palmitoleic acid	0.00022
L-Cysteine	0.00410	Octadecanoic acid	0.00035
L-Aspartate	0.02508	Oleic acid	0.00163
L-Asparagine	0.02841	Linoleate	0.00054
L-Arginine	0.02203	Linolenate	0.00008
L-Alanine	0.01334	Triacylglycerol	0.00573
Glycine	0.01077	Monoacylglycerol	0.00620
L-Glutamine	0.02158	Diacylglycerol	0.00087
		Sterol esters	0.01177
**Carbohydrates**			
Chitin	0.01368	**Soluble Pool**	
Mannan	0.14669	Thiamine	0.00290
β (1.3)-Glucan	0.23962	Ubiquinone-6	0.00290
		NADP+	0.00290
**Deoxyribonucleotides**		NAD+	0.00290
dTTP	0.02072	FMN	0.00290
dGTP	0.01266	FAD	0.00290
dCTP	0.01118	CoA	0.00290
dATP	0.02114	Biotin	0.00290
		Pyridoxal phosphate	0.00290
**Ribonucleotides**		5-Methyltetrahydrofolate	0.00290
UTP	0.00603		
GTP	0.00714		
CTP	0.00561		
ATP	0.00714		

**Table 3 jof-06-00171-t003:** Comparison between in vivo and in silico phenotypic behavior of *C. albicans* under different carbon and nitrogen sources. Growth (+); lack of growth (−).

	Biomass		
	In Vivo	In Silico	Reference
Carbon Source			
*N*-acetylglucosamine	+	+	[62,63]
Glucose	+	+	[61,62,63]
Maltose	+	+	[63]
Galactose	+	+	[61,62,63]
Sucrose	+	+	[63]
Fructose	+	+	[61,62,63]
Mannitol	+	−	This study
Acetate	+	+	[63]
Ethanol	+	+	[63]
Glycerol	+	+	[61,62,63]
Mannose	+	+	[61,62]
Citrate	+	+	[60]
Lactate	+	+	[62]
Sorbitol	+	+	[62]
L-sorbose	+	+	[60]
D-xylose	+	+	[60]
L-rhamnose	−	−	[60]
α,α-trehalose	+	+	[60]
Cellobiose	+	+	This study
Salicin	−	−	[60]
Myo-inositol	−	−	[60]
D-ribose	+	+	This study
Ribitol	−	−	[60]
D-glucuronate	−	−	[60]
D-galacturonate	−	−	[60]
Succinate	+	+	[60]
D-gluconate	+	+	[60]
Arbutin	−	−	[60]
D-arabinose	−	−	[60]
Galactitol	−	−	[60]
Starch	+	+	[60]
D-glucosamine	+	+	[60]
Inulin	−	−	[60]
Melibiose	−	−	[60]
Lactose	−	−	[60]
Raffinose	−	−	[60]
Erythritol	−	−	[60]
Xylitol	+	+	[60]
L-arabinitol	−	−	[60]
**Nitrogen Source**			
Nitrate	−	−	[60,64]
Nitrite	−	−	[60,64]
Ethylamine	+	−	[60]
L-Lysine	+	+	[60]
Ammonia	+	+	[60,64]
Cadaverine	+	−	[60]
Glucosamine	−	+	[60]
Creatine	−	−	[60]
Creatinine	−	−	[60]
Imidazole	−	−	[60]
L-asparagine	+	+	[60,64]
Urea	+	+	[60,64]
Hydroxylamine	−	−	[60,64]
Hydrazine	−	−	[60,64]
D-Tryptophan	−	−	[60]

**Table 4 jof-06-00171-t004:** Growth parameters of iRV781 and comparison with in vivo values for *C. albicans* and *S. cerevisiae*.

	Specific Growth Rate (h^−1^)	q (mmol g^−1^ dry weight h^−1^)			
		Glucose	Ethanol	Glycerol	Acetic Acid
In silico *C. albicans*	0.53	7.56	0	0	0
In vivo *C. albicans* [60]	0.51	7.56	0.38	0	0
In vivo *S. cerevisiae* [60]	0.38	13.26	21.87	1.98	<0.1

**Table 5 jof-06-00171-t005:** Anaerobic growth assessment of iRV781 model in defined media with or without anaerobic supplements. DMM [68] (defined minimal medium); DMM_sup._ [68] (defined minimal medium supplemented with ergosterol and Tween 80); GPP [67] (glucose-phosphate-proline); GPP_sup._ [67] (glucose-phosphate-proline supplemented with oleic acid and nicotinate).

Condition	Specific Growth Rate (h^−1^)	q (mmol g^−1^ dry weight h^−1^)		
		Glucose	Ethanol	Glycerol
In silico GPP	0	0	0	0
In silico GPP_sup._	0.08	6.58	10.80	0
In silico DMM	0	0	0	0
In silico DMM_sup._	0.08	6.58	10.80	0
*S. cerevisiae* DMM [68]	0.10	6.58	9.47	1.11

**Table 6 jof-06-00171-t006:** Drug targets evaluated for gene essentiality prediction in RPMI medium, as identified by the iRV781. Data retrieved from DrugBank database; only drugs with known pharmacological action were selected.

Systematic Name	Standard Name	EC Number	Organism	Drug	PDB Entry	Similarity	Coverage
C1_08590C_A	ERG1	1.14.14.17	*Candida albicans*	Terbinafine	-	-	-
			*Candida albicans*	Tolnaftate	-	-	-
C1_09720W_A	URA1	1.3.5.2	*Plasmodium falciparum*	Atovaquone	5DEL	37%	81%
C2_02460W_A	ERG7	5.4.99.7	*Candida albicans*	Oxiconazole	-	-	-
C5_00190C_A	FAS1	1.3.1.9	*Mycobacterium tuberculosis*	Ethionamide	4V8W	30%	45%
			*Mycobacterium tuberculosis*	Isoniazid			
C5_00770C_A	FOL1	4.1.2.25	*Saccharomyces cerevisiae*	Sulfacetamide	2BMB	42%	65%
C5_02710W_A	TRR1	1.8.1.9	*Staphylococcus aureus*	Azelaic acid	4GCM	42%	98%
C7_03130C_A	DFR1	1.5.1.3	*Escherichia coli*	Trimethoprim	4GH8	35%	77%
C5_00770C_A	FOL1	2.5.1.15	*Escherichia coli*	Sulfonamides and sulfones	1AJ2	36%	40%
			*P. falciparum*	Sulfonamides and sulfones	6KCM	26%	65%
C1_02420C_A	GSC1	2.4.1.34	*Candida albicans*	Anidulafungin	-	-	-
C1_05600W_A	GSL1		*Candida albicans*	Caspofungin	-	-	-
CR_00850C_A	GSL2		*Candida albicans*	Micafungin	-	-	-
C3_04830C_A	FAS2	2.3.1.41	*Escherichia coli*	Cerulenin	2BYX	31%	8%
CR_00850C_A	ERG11	1.14.14.154	*Candida albicans*	Azoles	-	-	-

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
