# Peer review of "Genome-Scale Metabolic Model of the Human Pathogen Candida albicans: A Promising Platform for Drug Target Prediction"

_jof, 2020, doi:10.3390/jof6030171_

Round 1

Reviewer 1 Report

The manuscript by Viana et al. describes a genome-scale metabolic model for the fungal pathogen Candida albicans that can be used in metabolic engineering platforms.

The comparison of the model prediction with experimental data available in the literature in terms of utilization of different nitrogen and carbon sources and growth in anaerobic conditions shows that the accuracy is comparable with similar models developed for other yeasts, like C. glabrata and S. cerevisiae. This study also explores the possibility to use this platform to facilitate the identification of potential drug targets, which is an important application especially considering the relative paucity of antifungal drugs licenced to treat Candida infection.

Major comments:

  • I could not access the supplementary material using the link that is provided at the end of the manuscript. As a result, I cannot comment on that. This is disappointing, as I believe it would have helped my understanding of the manuscript.
  • I found the methods section extremely concise. Although the authors in most cases provide the necessary references to back up the experimental design, pulling up the relevant papers each time breaks the reading flow. It would help the reader a lot if the “big picture” for each paragraph was explained a bit more (e.g. the method section from this relevant paper – provided in the References - was easier to follow: PMID: 24777859).
  • The authors refer to the Candida albicans strain that they used as reference strain SC5315. I believe that this is actually strain SC5314, and indeed the genome assembly available with accession number ASM18296v3 is the one of the SC5314 strain. This should be corrected throughout the manuscript.
  • The model is for Candida albicans. However, the authors themselves stress that multidrug-resistant clinical isolates are particularly common in the case of emerging non-albicans Candida species. It would be interesting to comment, in the discussion, on how easily (or not) the findings inferred from this model could be applied to related non-albicans species.
  • It would be interesting to have information about the distribution of the 781 genes among the 4 compartments considered by the authors (page 5, lines 155-157).
  • Page 6, lines 167-172. These lines could be rephrased to be clearer. My understanding is that, if we consider the proteins that have a shared EC number among the three models, more than 70% are in common. But then, how come that the next sentence reads that only about 65% of the proteins are shared by the three models?
  • Page 6; Lines 175-190: It would be interesting to expand the discussion of the potentially unique features in albicans. Is there anything available in the literature about those reactions in C. albicans, or maybe related non albicans species? The only comment is about enzyme 1.1.1.289: what about the others?
  • I think that paragraph 3.1.1 could provide more details. However, these could be in the File S2 that I cannot access.
  • Page 9: The comparison of the model with experimental data existing in the literature regarding the ability to use different nitrogen and carbon sources highlights that there is some degree of variability among different strains. However, this paragraph is confusing. Lines 231-232 read: “ CBS562 seems not to be able to use cellobiose and D ribose, contrary to the model’s prediction” (so I am inferring that the model predicts that albicans should grow on these carbon sources). However, a few lines later (lines 237-238), the author say “the results confirmed the model’s prediction regarding the utilization of cellobiose and D-ribose”, referring to the fact that SC5314 could not grow in these conditions (suggesting that the model correctly predicted that C. albicans could not grow). This seems to be in contrast with both what said above, and the information reported in Table 3 (where the model seems to predict growth in these conditions). Could the authors explain?
  • Do the authors think that the failure of the model to predict growth/absence of growth in the presence of Ethylamine, cadaverine, and glucosamine is due to the fact that strains other than SC5314 were used?
  • Page 12; line 335: “Within this new list”: where is this list?

Minor comments/typos:

  • There are some typos throughout the manuscript. Some examples are:
  • The name “merlin” is sometimes in italics, sometime not.
  • Page 4; Line 109: species name should be in italics
  • Page 4; Line 130: species name should be in italics
  • Page 5; Line 135: period missing
  • Page 5; Lines 162-163. Should the name of the model precede or follow the species name? Either way, it should be the same in the Table caption.
  • Page 6; Line 171: figure 2 should have capital letter
  • It would be easier if the nomenclature for the supplementary material were more consistent (e.g. “Supplementary file III” versus “file S2”)
  • Page 10; Line 259: flux was fixed o > flux was fixed to
  • Table 4: I suggest adding the species name in the lines of the table for clarity (In silico C. albicans, In vivo C. albicans, In vivo S. cerevisiae).
  • Page 12; Lines 303-304: I am not sure I understand what the authors mean
  • Page 13; Table 6: the “pharmacological action” column s redundant, as all the rows are “yes”, and the legend already states “only drugs with known pharmacological action were selected”. Also, species name should be in extenso.

Author Response

Thank you for your detailed review of our manuscript. We have tried to accommodate all your suggestions in the revised version, which we feel improved significantly thanks to your contribution.

Please notice bellow the point-by-point response to the raised concerns:

Major comments:

  • I could not access the supplementary material using the link that is provided at the end of the manuscript. As a result, I cannot comment on that. This is disappointing, as I believe it would have helped my understanding of the manuscript.

DUE TO A MISUNDERSTANDING DURING THE SUBMISSION PROCESS, WE DID NOT UPLOAD THE SUPPLEMENTARY FILES. WE APOLOGIZE. THEY ARE AVAILABLE IN THE CURRENT SUBMISSION.

  • I found the methods section extremely concise. Although the authors in most cases provide the necessary references to back up the experimental design, pulling up the relevant papers each time breaks the reading flow. It would help the reader a lot if the “big picture” for each paragraph was explained a bit more (e.g. the method section from this relevant paper – provided in the References - was easier to follow: PMID: 24777859).

THE METHODS SECTION WAS MADE MORE COMPREHENSIVE.

  • The authors refer to the Candida albicansstrain that they used as reference strain SC5315. I believe that this is actually strain SC5314, and indeed the genome assembly available with accession number ASM18296v3 is the one of the SC5314 strain. This should be corrected throughout the manuscript.

IN FACT, DUE TO A FORMATTING ERROR THE NUMBER 4 HAD BEEN CHANGED TO THE 5, FOR DISTRACTION IT WENT UNNOTICED, THANKS FOR NOTICING. THE MANUSCRIPT WAS CORRECTED ACCORDINGLY.

  • The model is for Candida albicans. However, the authors themselves stress that multidrug-resistant clinical isolates are particularly common in the case of emerging non-albicans Candida species. It would be interesting to comment, in the discussion, on how easily (or not) the findings inferred from this model could be applied to related non-albicans species.

WE ANALYZED THE EXISTENCE OF ORTHOLOGS FOR THE ENZYMES PRESENT IN TABLE 6 IN SOME RELEVANT PATHOGENIC CANDIDA SPECIES. NAMELY C. PARAPSILOSIS, C. DUBLINIENSIS, C. AURIS AND C. GLABRATA. IT ALLOWED TO VERIFY THAT DUE TO THE PROXIMITY BETWEEN THE SPECIES, IT MAY BE POSSIBLE TO EXPAND THE ANALYSIS OF THE MODEL TO OTHER CANDIDA SPECIES. THIS INFORMATION HAS BEEN ADDED TO THE DISCUSSION SECTION.

  • It would be interesting to have information about the distribution of the 781 genes among the 4 compartments considered by the authors (page 5, lines 155-157).

THIS INFORMATION HAS BEEN INCLUDED.

  • Page 6, lines 167-172. These lines could be rephrased to be clearer. My understanding is that, if we consider the proteins that have a shared EC number among the three models, more than 70% are in common. But then, how come that the next sentence reads that only about 65% of the proteins are shared by the three models?

IT IS TRUE THAT THE SENTENCE COULD BE CONFUSING, IT WAS REFORMULATED IN ORDER TO BE CLEARER.

  • Page 6; Lines 175-190: It would be interesting to expand the discussion of the potentially unique features in albicans. Is there anything available in the literature about those reactions in  albicans, or maybe related non albicans species? The only comment is about enzyme 1.1.1.289: what about the others?

INFORMATION REGARDING THE REMAINING ENZYMES WAS ADDED.

  • I think that paragraph 3.1.1 could provide more details. However, these could be in the File S2 that I cannot access.

INDEED, THE INFORMATION PROVIDED IN THE SUPPLEMENTARY MATERIAL (NOW ACCESSIBLE) COMPLEMENTS THAT OF PARAGRAPH 3.1.1.

  • Page 9: The comparison of the model with experimental data existing in the literature regarding the ability to use different nitrogen and carbon sources highlights that there is some degree of variability among different strains. However, this paragraph is confusing. Lines 231-232 read: “ CBS562 seems not to be able to use cellobiose and D ribose, contrary to the model’s prediction” (so I am inferring that the model predicts that albicansshould grow on these carbon sources). However, a few lines later (lines 237-238), the author say “the results confirmed the model’s prediction regarding the utilization of cellobiose and D-ribose”, referring to the fact that SC5314 could not grow in these conditions (suggesting that the model correctly predicted that  albicanscould not grow). This seems to be in contrast with both what said above, and the information reported in Table 3 (where the model seems to predict growth in these conditions). Could the authors explain?

CELLOBIOSE AND D-RIBOSE:

  1. ALBICANS CBS562 STRAIN IS NOT TO BE ABLE TO USE CELLOBIOSE AND D-RIBOSE ACCORDING TO LITERATURE. HOWEVER, OUR MODEL PREDICTS GROWTH IN THE PRESENCE OF THESE CARBON SOURCES. WE CONFIRMED EXPERIMENTALLY THAT, IN FACT, C. ALBICANS SC5314 IS ABLE TO USE CELLOBIOSE AND D-RIBOSE, AND THUS THAT THE IN SILICO PREDICTION WAS CORRECT (IN THE TABLE, WE PREFERABLY INCLUDE DATA RELATED TO THE REFERENCE STRAIN, IF AVAILABLE).

MANNITOL (DIFFERENT CASE):

ACCORDING TO LITERATURE C. ALBICANS IS ABLE TO USE MANNITOL AS CARBON SOURCE. HOWEVER, OUR MODEL DID NOT PREDICT GROWTH USING MANNITOL AS CARBON SOURCE.

WE CONFIRMED EXPERIMENTALLY THAT C. ALBICANS SC5314 IS INDEED ABLE TO USE MANNITOL AS CARBON SOURCE. SO, THE IN SILICO PREDICTION WAS WRONG. 

THE TEXT WAS REWRITEN FOR THE SAKE OF CLARITY.

  • Do the authors think that the failure of the model to predict growth/absence of growth in the presence of Ethylamine, cadaverine, and glucosamine is due to the fact that strains other than SC5314 were used?

DUE TO LACK OF REAGENTS, WE WERE UNABLE TO TEST THESE NITROGEN SOURCES IN OUR LABORATORY. SINCE SOME CARBON SOURCES WERE CORRECTED AFTER THE TESTS WE PERFORMED, THE SAME IS LIKELY TO HAPPEN WITH NITROGEN SOURCES IF WE WERE ABLE TO TEST THEM. FOR EXAMPLE, IN THE CASE OF D-GLUCOSAMINE, GIVEN THE PRESENCE OF THE ENZYMES MENTIONED BEFORE, IT IS LIKELY THAT C. ALBICANS IS ALSO CAPABLE OF USING IT AS A NITROGEN SOURCE. THIS NOTION WAS FURTHER DISCUSSED IN THE MANUSCRIPT.

  • Page 12; line 335: “Within this new list”: where is this list?

THERE IS NO “NEW LIST”. THIS SENTENCE WAS THE REMNANT OF A PRIOR VERSION OF THE MANUSCRIPT. IT HAS BEEN CORRECTED.

Minor comments/typos:

  • There are some typos throughout the manuscript. Some examples are:
  • The name “merlin” is sometimes in italics, sometime not.

CORRECTED, ALL ITALICS

  • Page 4; Line 109: species name should be in italics

CORRECTED.

  • Page 4; Line 130: species name should be in italics

CORRECTED

  • Page 5; Line 135: period missing

CORRECTED

  • Page 5; Lines 162-163. Should the name of the model precede or follow the species name? Either way, it should be the same in the Table caption.

ADJUSTED

  • Page 6; Line 171: figure 2 should have capital letter

DONE

  • It would be easier if the nomenclature for the supplementary material were more consistent (e.g. “Supplementary file III” versus “file S2”)

“SUPPLEMENTARY FILE III” WAS CHANGED TO “FILE S3”.THE NOMENCLATURE "FILE S.." WAS USED IN ALL THE CASES, ACCORDING TO THE GUIDELINES PROVIDED.

  • Page 10; Line 259: flux was fixed o > flux was fixed to

DONE!

  • Table 4: I suggest adding the species name in the lines of the table for clarity (In silico C. albicans, In vivo C. albicans, In vivo S. cerevisiae).

ADJUSTED

  • Page 12; Lines 303-304: I am not sure I understand what the authors mean

YOU ARE RIGHT. THAT SENTENCE DID NOT MAKE MUCH SENSE, SO WE DECIDE TO REMOVE IT.

  • Page 13; Table 6: the “pharmacological action” column s redundant, as all the rows are “yes”, and the legend already states “only drugs with known pharmacological action were selected”. Also, species name should be in extenso.

“PHARMACOLOGICAL ACTION” COLUMN REMOVED. SPECIES NAME ARE NOW IN EXTENSO.

Reviewer 2 Report

In this paper, the authors report a first genome-scale metabolic model of Candida albicans. C. albicans is one of the most important human fungal pathogens. It is timely and important to develop a system level characterization of its metabolic network. The authors used standard modeling methods to reconstruct the global metabolic network. It is commendable that the authors experimentally validated several predicted pathways involved in carbon and nitrogen utilization.

 Major Comments:

  1. The reconstructed albicans metabolic network does not seem to capture key C. albicans specific features. The comparison with the networks of baker’s yeast and C. glabrata showed that 20% of proteins were unique in the C. albicans model iRV781. These proteins deserve specific consideration and investigation. The manuscript only includes a list of the enzymes (line 175-188), without detailed discussion and investigation of their potential function and implications in pathogenesis, virulence, or morphogenesis. An enormous amount of information has been collected in the Candida Genome Database. The dynamic expression of genes involved in metabolic pathways and their regulation will provide important insights into biology of C. albicans. Such in-depth investigation appears missing.
  2. The network was built upon KEGG, metacyc, BRENDA, TCDB, etc., with flux-balance analysis using biomass estimation as described previously in Reference 37. It is unclear what literature or experimental data were used for the analysis of albicans. The same theoretical ratio used in S. cerevisae was used for the phosphorus to oxygen ratio. It is not described what the rationale was.   
  3. The key claim of the manuscript, that this metabolic network provides a platform for drug prediction, appears to be an overstatement. First of all, the details for essentiality analysis is completely missing in the Methods section. What was the environmental condition for simulations? What were the parameters for predicting essentiality? How to evaluate the confidence of the drug target prediction? Secondly, the section 3.3. which includes Table 6 and a list of drug targets, appears speculative. Most of the results described from Line 305-345 were indeed discussion. At the minimum, the accession number and gene name for each predicted target gene (which encodes the enzymes being targeted) should be provided. It is important to examine the level of similarity between the C. albicans genes and the homologs in the organism with known drugs. The column of “pharmacological action” is misleading, in the context of C. albicans networks.
  4. A phylogenetic tree was inferred using MEGA X, but details of the inference methods were not described. Was it inferred based on maximum likelihood, neighbor-joining or other method? What were the evolutionary model parameters? In addition, it is suggested to perform Bootstrap tests to test the statistical significance of branching points.
  5. The manuscript perhaps will benefit from including a Discussion section, which discusses in detail the strengths and potential limitations of the reconstructed models.

Minor Comments:

  1. The manuscript perhaps will benefit from proofreading and English editing.
  2. Please italicize species names.
  3. Figure legends are missing for Supplementary Figure 1.
  4. Two strains of albicans are included in Supplementary Figure 1. What strain is it associated with accession number E15168.1?

Author Response

Thank you for your detailed review of our manuscript. We have tried to accommodate all your suggestions in the revised version, which we feel improved significantly thanks to your contribution.

Please notice bellow the point-by-point response to the raised concerns:

 Major Comments:

  1. The reconstructed albicans metabolic network does not seem to capture key C. albicans specific features. The comparison with the networks of baker’s yeast and C. glabrata showed that 20% of proteins were unique in the C. albicans model iRV781. These proteins deserve specific consideration and investigation. The manuscript only includes a list of the enzymes (line 175-188), without detailed discussion and investigation of their potential function and implications in pathogenesis, virulence, or morphogenesis. An enormous amount of information has been collected in the Candida Genome Database. The dynamic expression of genes involved in metabolic pathways and their regulation will provide important insights into biology of C. albicans. Such in-depth investigation appears missing.

INFORMATION REGARDING THE REMAINING ENZYMES WAS ADDED.

  1. The network was built upon KEGG, metacyc, BRENDA, TCDB, etc., with flux-balance analysis using biomass estimation as described previously in Reference 37. It is unclear what literature or experimental data were used for the analysis of albicans. The same theoretical ratio used in S. cerevisae was used for the phosphorus to oxygen ratio. It is not described what the rationale was.  

THE LITERATURE DATA USED FOR BIOMASS ESTIMATION IS DESCRIBED IN THE SECTION 3.1.2, ADDITIONALLY IN FILE S1 THE INFORMATION ABOUT BIOMASS COMPOSITION IS MORE DETAILED. PHOSPHORUS TO OXYGEN RATIO WAS BETTER EXPLAINED.

  1. The key claim of the manuscript, that this metabolic network provides a platform for drug prediction, appears to be an overstatement. First of all, the details for essentiality analysis is completely missing in the Methods section. What was the environmental condition for simulations? What were the parameters for predicting essentiality? How to evaluate the confidence of the drug target prediction? Secondly, the section 3.3. which includes Table 6 and a list of drug targets, appears speculative. Most of the results described from Line 305-345 were indeed discussion. At the minimum, the accession number and gene name for each predicted target gene (which encodes the enzymes being targeted) should be provided. It is important to examine the level of similarity between the C. albicans genes and the homologs in the organism with known drugs. The column of “pharmacological action” is misleading, in the context of C. albicans networks.

DETAILS REGARDING GENE ESSENTIALITY SIMULATIONS HAVE BEEN ADDED TO THE METHODS. A SET OF ESSENTIAL GENES CONFIRMED BY LITERATURE WERE USED TO ASSESS THE MODEL'S CAPACITY TO PREDICT ESSENTIALITY. INFORMATION RELATED TO GENES HAS BEEN ADDED TO TABLE 6.

IN THE FIRST VERSION, WE DECIDED NOT TO INCLUDE THE PERCENTAGE OF SIMILARITY BETWEEN DE PROTEINS, SINCE OUR IDEA IS TO SHOW THAT THE MODEL WILL BE ABLE TO CORRECTLY PREDICT ESSENTIAL ENZYMES / GENES IN THE CONDITIONS THAT MICROORGANISMS FACE INSIDE HUMAN SERUM, ENZYMES THAT CAN BE POTENTIAL DRUG TARGETS BUT NOT NECESSARILY THE SAME THAT WE PRESENT IN THE TABLE,  AND NOT NECESSARILY TARGETS OF THE SAME DRUG. HOWEVER, WE ACCEPT THAT IT MAY BE A RELEVANT INFORMATION TO INCLUDE. SIMILARITY BETWEEN THE C. ALBICANS ENZYMES AND THE ENZYMES IN THE TARGET ORGANISM WAS ADDED AS A COLUMN TO THE TABLE. 

 “PHARMACOLOGICAL ACTION” COLUMN WAS REMOVED FROM TABLE 6.

  1. A phylogenetic tree was inferred using MEGA X, but details of the inference methods were not described. Was it inferred based on maximum likelihood, neighbor-joining or other method? What were the evolutionary model parameters? In addition, it is suggested to perform Bootstrap tests to test the statistical significance of branching points.

 MORE DETAILED INFORMATION WAS ADDED. PLEASE SEE THE LEGEND OF FIGURE S1.

  1. The manuscript perhaps will benefit from including a Discussion section, which discusses in detail the strengths and potential limitations of the reconstructed models.

ALTHOUGH WE DID NOT SPLIT THE RESULTS AND DISCUSSION SECTION INTO TO TWO, TWO EXTRA PARAGRAPHS WERE ADDED TO INCLUDE THE SUGGESTED DISCUSSION.

 Minor Comments:

  1. The manuscript perhaps will benefit from proofreading and English editing.

THE MANUSCRIPT WAS REVIEWED.

  1. Please italicize species names.

DONE!

  1. Figure legends are missing for Supplementary Figure 1.

Two strains of albicans are included in Supplementary Figure 1. What strain is it associated with accession number E15168.1?

LEGENDS WERE INCLUDED. IN FACT, BY MISTAKE, THE WRONG FIGURE HAD BEEN UPLOADED IN THE ANNEXES. PLEASE CHECK THE CURRENT CORRECT FIGURE S1.